# Emerging Targets for the Treatment of Osteoarthritis: New Investigational Methods to Identify Neo-Vessels as Possible Targets for Embolization

**DOI:** 10.3390/diagnostics12061403

**Published:** 2022-06-06

**Authors:** Reza Talaie, Pooya Torkian, Alexander Clayton, Stephanie Wallace, Hoiwan Cheung, Majid Chalian, Jafar Golzarian

**Affiliations:** 1Vascular and Interventional Radiology, Department of Radiology, University of Minnesota, Minneapolis, MN 55455, USA; rtalaie@umn.edu (R.T.); aclayton@umn.edu (A.C.); walla649@umn.edu (S.W.); jafar@umn.edu (J.G.); 2Department of Radiology, Division Musculoskeletal Imaging and Intervention, University of Washington, Seattle, WA 98195, USA; hcheung@uw.edu (H.C.); mchalian@uw.edu (M.C.)

**Keywords:** osteoarthritis, genicular artery embolization, embolization

## Abstract

Osteoarthritis (OA) is the major cause of disability, affecting over 30 million US adults. Continued research into the role of neovascularization and inflammation related to osteoarthritis in large-animal models and human clinical trials is paramount. Recent literature on the pathogenetic model of OA has refocused on low-level inflammation, resulting in joint remodeling. As a result, this has redirected osteoarthritis research toward limiting or treating joint changes associated with persistent synovitis. The overall goal of this review is to better understand the cellular and tissue-specific mechanisms of inflammation in relation to a novel OA treatment modality, Genicular Artery Embolization (GAE). This article also assesses the utility and mechanism of periarticular neovascular embolization for the treatment of OA with a particular emphasis on the balance between pro-angiogenic and anti-angiogenic cytokines, inflammatory biomarkers, and imaging changes.

## 1. Introduction

Traditionally considered a “wear and tear” phenomena of the bone and cartilage, osteoarthritis (OA) is increasingly understood to represent sequela of chronic inflammatory processes [1]. As our basic understanding of osteoarthritis pathology broadens, so too do potential treatment targets. Prior literature has identified that the neovascularization of joint tissues plays a significant role in the pathology of OA and is a suggested target for future treatment [2]. Continued research into the role of neovascularization and inflammation related to osteoarthritis in large-animal models and human clinical trials is paramount. Recent literature has shifted the pathogenetic model of OA to refocus on low-level inflammation, resulting in joint remodeling [3]. Chronic inflammation alters chondrocyte function, shifting normal cell signaling to pro-inflammatory cytokines, which in turn promotes angiogenesis [3]. Multiple small animal models have demonstrated that the degree of angiogenesis correlates with more severe OA [4,5,6,7,8,9,10,11]. These new vessels may in turn function as a conduit for continued joint inflammation and new neuronal migration [2]. These nerves are sensitized to pain due to their subjection to hypoxia, inflammation, and mechanical stress within the joint [12]. This new understanding of the pathophysiology of OA has served as a target opportunity for new treatment modalities to address gaps in clinical needs. Development in the pathogenetic model of OA has redirected research in its treatment and shifted the focus to limiting or treating joint changes associated with persistent synovitis. 

Okuno et al. have employed targeted neo-vessel embolization to successfully treat symptomatic osteoarthritis with durable therapeutic response [13]. Early results of geniculate artery embolization (GAE) demonstrate improved patient pain and function [14,15,16,17,18,19,20]. GAE is hypothesized to limit inflammation and pain in OA via embolization of neo-vessels. Treatments of neovascularization in osteoarthritis show promise with GAE but will require robust randomized clinical trials before the utility of this procedure can be fully established. The overall goal of this review is to better understand the cellular and tissue-specific mechanisms of inflammation in relation to a novel OA treatment modality, GAE. We assess the utility and mechanism of periarticular neovascular embolization for the treatment of OA with a particular emphasis on understanding the balance between the pro and anti-angiogenic cytokines, inflammatory biomarkers, and imaging changes. 

## 2. Global Prevalence, Natural History, Risk Factors, Pathophysiology, and Treatment of OA

### 2.1. Epidemiology and Economic Burden of OA 

Of all types of arthritis, osteoarthritis (OA) is the most common, affecting over 30 million US adults [21]. The radiographic incidence of symptomatic knee OA has an incidence of 4.3% in men and 8.1% in women of all ages. It has been shown that the overall prevalence of radiographic OA reaches 37.1% [22] with nearly half of people with symptomatic OA experiencing physical disability. As the population ages, the incidence, prevalence, and economic burden of its treatment and disability will increase [23]. In the United States, the number of individuals over the age of 65 is projected to rise to 78 million by 2035 from 49.2 million in 2016 [24]. The prevalence of total knee arthroplasty (TKA) was demonstrated to be 4.7 million individuals in 2010 with the overall trend being of increasing prevalence over time [25]. This comes at no small cost, with the annual total hospitalization charges for TKA nearly quadrupling from $8.1 billion in 1998 to $38.5 billion in 2011 [26]. The average lifetime direct medical cost for treatment of those diagnosed with OA is estimated to be $12,400 or 10% of all estimated direct medical expenses for those individuals. Most of those costs affect the 54% of OA patients who undergo TKA, which on average costs $20,293, and for patients who require revision surgery, resulting in additional costs averaging $29,388. Non-surgical regimens are estimated between $494 and $684 annually [27]. The skyrocketing costs of an aging population coupled with expanded TKA eligibility have led Losina et al. to conclude that there is legitimate need for more effective non-operative therapies [27]. 

### 2.2. Natural History, Prognosis, and Imaging of OA

Patients with knee OA can present with joint pain, stiffness, bony crepitus, joint edema, and physical deformities. Radiographic findings include narrowing of the joint compartment, osteophytes, and subchondral sclerosis [28]. Current non-operative treatment is limited to physical therapy, oral anti-inflammatories, and intra-articular corticosteroid/hyaluronic acid injections. Non-steroidal anti-inflammatories (NSAIDs) are well tolerated but are not without risk, as they may cause acute renal failure, gastritis, or interfere with platelet aggregation [29]. Arthroplasty comes with its own risks of perioperative morbidity and mortality [30,31]; hence, this is reserved for severe OA, resulting in significant lifestyle limitations.

### 2.3. Pathophysiology: Understanding of the Balance between Pro and Anti-Angiogenic Cytokines, Inflammatory Biomarkers

While the pathophysiology of knee osteoarthritis has traditionally been described in the context of inflammation and joint space narrowing, new evidence in the literature suggests that abnormal blood vessel formation may also play a role. Osteoarthritic pain is thought to be caused by joint space inflammation, abnormal innervation of synovial structures and increased sensitization of the central and peripheral nervous systems (Figure 1).

Synovitis may damage underlying tissues by altering chondrocyte function, while increased angiogenesis and bony remodeling contribute to chronic inflammation in OA [1]. Angiogenesis is the process by which new capillaries are formed from pre-existing blood vessels and plays an important role in physiologic wound healing. Pathologic angiogenesis can result in chronic inflammatory conditions and the metastatic spread of tumors [1]. Vasculogenesis, a consequence of angiogenesis, occurs when circulating angioblasts differentiate into endothelial cells. Matrix metalloproteinases (MMP) and other cytokines establish an environment for the new arterioles and venules to form in the perivascular space [32]. Vascular endothelial growth factor (VEGF) and platelet-derived growth factor (PDGF) help maintain blood vessel stability after MMPs create new ostia by destroying part of the existing endothelium [1]. The process by which endothelial cells proliferate and localize to avascular spaces are known as “angiogenic sprouting” [3]. In osteoarthritis, neo-vessels may contribute to persistent inflammation by maintaining oxygen and nutrients to abnormal endothelial cells as well as giving pro-inflammatory cytokines access to the local microenvironment [3].

Recent attention has focused on the local microenvironment as it pertains to osteoarthritis. Like generic inflammation, regulatory molecules and cytokines function in a complex, sophisticated manner to stimulate and/or inhibit angiogenesis [1]. Compounds that upregulate angiogenesis include prostaglandin E2 (PGE2), histamine, VEGF, interleukin-1 (IL-1), PDGF, and nitric oxide (NO) [1]. VEGF is itself stimulated by pro-inflammatory cytokines, such as IL-1, interleukin 17 (IL-17), tumor necrosis factor alpha (TNF-alpha), NO, and reactive oxygen species [33]. Some factors that inhibit angiogenesis in the synovium include chondromodulin-1, interferon gamma (IFN-gamma), interleukin-4 (IL-4), and tissue inhibitors of MMP-1 and 2 [1]. Chondromodulin 1 and U-995 have been shown to not only discourage angiogenesis but to prevent endothelial cell production and migration [32]. Normally, pro-inflammatory and anti-inflammatory cytokines exist in a complex, homeostatic environment. In certain inflammatory conditions, the balance is tilted toward pro-inflammatory cytokines. While conventionally associated with other inflammatory diseases, this net catabolic context may exist in processes that have traditionally been understood as non-inflammatory, such as osteoarthritis [3].

Articular cartilage is an avascular tissue. The blood supply to the cartilage is derived from the adjacent, vascular synovium. Cartilage is a critical component of the joint, as it helps to evenly distribute high mechanical forces and maintain the structure of the joint as a unit [28]. Chondrocytes are specifically resistant to angiogenesis, as they secrete anti-angiogenic factors, such as troponin 1, chondromodulin 1, and matrix metalloproteinase inhibitors [33]. Normal cartilage is composed of type II collagen. In early osteoarthritis, there is loss of the proteoglycans and glycosaminoglycans that are normally found in the extracellular matrix of the chondrocytes. Type X collagen may be involved in the pathophysiology of endochondral ossification in knee OA. Type I collagen is also present in vascularized, osteoarthritic knee joint cartilage [34].

Nerve tissue tends to parallel vascular structures. The synovium and subchondral bone contain sensory nerves, which consist of unmyelinated C fibers. This nervous tissue functions along with the neuropeptides Substance P and calcitonin gene-related peptide [28]. Mechanical forces, hypoxia, and inflammation may increase nerve sensitivity to pain [12]. Hypoxic conditions stimulate the release of collagen prolyl 4-hydroxylase (P4Halpha(I)) and hypoxia-inducible factor (HIF-1 alpha) [35]. Normal cartilage is not innervated; however, with pathological vascularization, the perivascular tissue and cartilage may become innervated and more sensitive to pain [28]. Chondrocytes have been found to possess not only receptors for substance P and calcitonin generated peptide but also adrenergic receptors and vasoactive intestinal peptide (VIP) receptors. Peripheral nerve fibers are normally found in periosteal tissue and trabecular bone. These fibers are involved physiologically in normal developmental processes, such as endochondral ossification and limb formation. Additionally, they provide sympathetic and sensory innervation, which assists with healing fractures [36]. Pathologically, the sensory and sympathetic neurotransmitters may have a role in inflammatory arthropathies, such as rheumatoid arthritis [36]. Sensory and sympathetic nerve fibers are concentrated in human cartilage associated with tibiofemoral osteoarthritis and marginal osteophytes. The abnormal nerve fibers have been observed accompanying abnormal vasculature originating from subchondral bone, extending into the cartilaginous tissue [36].

### 2.4. A Review of Translational Animal Models

Larger animal model studies have shown angiogenic invasion to be statistically increased in OA knee joints across the osteochondral junction [8]. McDougall et al. showed increased angiogenesis in the medial collateral ligament (MCL) in induced OA joints compared to sham joints in rabbits [9]. Jansen et al. showed that following bilateral ACL transections in rabbits, VEGF was present in the joint cartilage but not present in the cartilage of control animals [10]. Bray et al. showed a five-fold increase in blood flow to the meniscus at four weeks post injury, which was statistically significant compared to control and sham-operated animals [11]. 

### 2.5. GAE: Anatomy of Genicular Arteries

GAE requires a fundamental knowledge of knee vascular anatomy, along with its variations, for effectively and safely identifying the embolizing target [37,38]. Three main vessels provide the vascular supply to the knee. The descending genicular artery (DGA) branching from the femoral artery supplies the superior knee, the anterior tibial recurrent artery (ATRA) branching from the anterior tibial artery supplies the inferior knee, and the genicular arteries arising from the popliteal artery supply the remainder of the knee joint [39]. Because the medial aspect of the joint is the weight-bearing portion of the knee, the medial knee joint compartment is more commonly affected by OA than the lateral compartment [37,40,41]. Consequently, the medial genicular branches of the popliteal artery and the DGA branches are of particular importance, as these vessels are common targets for GAE [37]. Several studies, however, have shown the high degree of variation in genicular vessels and DGA [39,40,42,43,44,45,46,47]. A recent cadaver study by Sighary et al. noted that although the most common genicular artery pattern was independent branches of the superior medial (SMGA), superior lateral (SLGA), inferior medial (IMGA), inferior lateral (ILGA), and medial genicular artery (MGA) from the popliteal artery, 72% of cadavers had genicular artery variations [39]. Most variants were related to the origin of the MGA, which is consistent with previous studies [48,49]. Additionally, the same cadaver study noted anatomical variations of the DGA and its muscular (MB), articular (AB), and saphenous branches (SB). Previous plastic surgery literature has created a DGA classification system of seven subtypes [47,50]. Shighary et al., however, utilized a three-subtype classification system of DGA that is more oriented toward GAE. In this, the MB was excluded because it can easily be identified during GAE and is not likely to be a site of nontarget embolization. In addition, the medial epicondyle and the origin of the DGA from the femoral artery were used as landmarks for classification and can be easily visualized during GAE. Type A classification was defined as the division of the AB and SB above the midpoint of the medial epicondyle and origin of the DGA, and Type B was defined as the division below this midpoint. Type C classification was defined as separate AB and SB origins from the femoral artery [39]. Results showed 72% of DGAs were classified as Type B, 24% were classified as Type A, and 4% were classified as Type C [39]. Angiographic studies also correlate with these cadaveric findings. In a recent study by Bagla et al., angiographic findings from 39 GAE procedures showed anatomical variations of the medial and lateral genicular arteries [38]. Three medial branches (DGA, SMGA, and IMGA) and three lateral branches (SLGA, ILGA, and ATRA) were analyzed. As opposed to previous classification systems that focused primarily on the MGA [39,42,43], Bagla et al. created a new classification system of genicular artery anatomy that excluded the MGA due to its limited perfusion of the knee. For the medial aspect of the knee, M1 was classified as the presence of all three medial branches, whereas M2 was classified as the presence of two of the three medial branches. For the lateral knee, the presence of all three lateral branches was classified as L1 while L2 was classified as the presence of two of the three. This provides a classification system clinically oriented toward consistent reporting for GAE and better predicting anastomotic networks [38]. 

Anastomotic networks between the genicular arteries provide additional complex variation to the vascular network of the knee. Bagla et al. observed anastomoses in 26.4% of genicular arteries, with highest rates between the musculoarticular branch of the DGA and the SMGA/SLGA [38]. Other cadaver studies have observed anastomoses between the DGA and SMGA and between the IMGA and medial sural artery [41]. Although the clinical impact of these anastomoses was not discussed, Little et al. noted the impact vessel anastomoses within the knee vasculature network can have on nontarget embolization [18]. In their prospective pilot study, three cases were reported to have significant retrograde flow through geniculate anastomoses from the target artery to the popliteal artery, increasing the risk of embolization of nontarget sites. As a result, these patients were not embolized [18]. Additionally, GAE poses a particular challenge for coil embolizing anastomotic vessels to prevent nontarget embolization. The knee joint, patella, distal femur, and proximal tibia receive blood from the geniculate arteries [51]. Combined GAE and coil embolization for nontarget embolization prevention could result in osteonecrosis, which would provide poor GAE outcomes and future osseous complications [18]. Additionally, the genicular arteries provide a collateral network of vessels in peripheral vascular disease [18,43]. 

Knowledge of knee vascular anatomy is essential for minimizing risk of GAE. A common side effect of GAE is skin discoloration secondary to cutaneous ischemia from nontarget embolization of cutaneous arteries. Okuno et al. reported that four patients experienced transient color change of the overlying skin of the treated knee that spontaneously resolved by 1 month follow-up [13]. Bagla et al. observed a similar adverse event after GAE, with 13 of 20 patients noting skin discoloration that self-resolved within 3 months of GAE [14]. O’Grady et al. and other studies have observed cutaneous supply from the DGA and SLGA [41,50,52]. Additionally, unavoidable cutaneous branches are present on the ILGA and IMGA [41]. Little et al. minimized the nontarget embolization of cutaneous arteries by utilizing ice packs to constrict cutaneous vessels temporarily [18]. As a result, the GENESIS study observed a much lower 12.5% [18] rate of skin discoloration compared to 65% [14] and 57% [13] in previous studies. Common origins are also important to consider for minimizing the risk of nontarget embolization. Common trunks involving the MGA are especially important, as nontarget embolization may result in damage to the cruciate ligaments [37,41,43,53]. O’Grady et al. observed three variants of vessels with a common origin of the MGA: the SMGA (5 out of 20 cadavers), SLGA (4 out of 20 cadavers), and both the SMGA and SLGA (1 out of 20) [41]. Sighary et al. observed similar results with 45 out of 196 cadavers having an SMGA and MGA common trunk, 31 out of 204 having an SLGA and MGA common trunk, and 20 out of 204 having an MGA common trunk with SLGA and MLGA [39]. MSGA and LSGA common trunks must also be considered, as nontarget embolization of the contralateral side could result [38]. Overall, a comprehensive knowledge of the knee vascular network is essential for GAE. Recognition of normal and variant anatomy as well as anastomotic connections can help minimize risks associated with GAE and reduce procedure time. 

### 2.6. A Continued Need for OA Treatment

Advancements in the pathogenetic model of OA has redirected research into its treatment and shifted the focus to limiting or treating joint changes associated with persistent synovitis. Okuno et al. has published four studies in human subjects showing that targeted neovessel embolization successfully treated symptomatic osteoarthritis with durable therapeutic response. Twenty-five patients with radiographic and clinical findings of knee osteoarthritis underwent the angiography and embolization of identifiable neo-vessels. These vessels tended to be associated with the synovium, infrapatellar fat pad, medial meniscus, medial joint capsule, and the periosteum adjacent to the medial condyle. They were identified as “excessive, disorganized” vascular structures, which often demonstrated arteriovenous shunting and early venous drainage [16]. Neo-vessels were embolized using 10–70 micrometer imipenem/cilastatin (IPM/CS) particles or 75 micrometer microspheres loaded with the same drug. Patients reported substantial pain relief and improvement in symptoms after embolization. Interestingly, some patients experienced clinical improvement minutes after the procedure, whereas others reported clinical improvement several weeks to months after treatment [17]. 

Continued follow-up with magnetic resonance imaging in treated patients demonstrated significantly improved synovitis. These results suggest that embolization may function to treat pain and modify disease progression. While the clinical results are promising, the effect of embolization on laboratory, histological and imaging findings of OA are not well understood. Imaging studies have similarly coalesced on the association between OA and synovitis. Separate from any study related to GAE, but speaking more broadly of synovitis, Macfarlane et al. recently stated, “changes in synovitis, whether persistently extensive or intermittent, are associated with cartilage damage over time.” They ultimately conclude, “Since synovitis is a potentially modifiable intraarticular feature, further research is warranted to assess whether treatment of synovitis mitigates cartilage destruction” [54]. Results from the Multicenter Osteoarthritis Study (MOST) have created compelling evidence of synovitis as an independent cause of OA as well as a potential modifiable contributor to the disease [55]. As it relates to pain, there is early evidence that there is a direct correlation between the Western Ontario and McMaster Universities Osteoarthritis Index (WOMAC) pain scale and synovitis [56]. 

## 3. New Opportunities in the Treatment of OA

There remains a sizable portion of knee OA patients who do not respond to non-operative therapy and are not considered good surgical candidates due to associated comorbidities or early disease stage. Therapeutic embolization for osteoarthritis has had promising early results [14,15,16,17,18,19,20,57,58]. Heller et al. reviewed the technical success, defined as embolization of at least one target genicular artery, and clinical success of GAE [37]. According to Heller et al., the technical success of GAE has been reported to range from 84 to 100%, and GAE has shown clinical success based on WOMAC [13,14], KOOS [15,18], and VAS scores [13,14,18,57]. A recent meta-analysis by Torkian et al. also validated the therapeutic success of GAE on OA [59]. In the 11 included studies, GAE resulted in significant improvement of VAS and WOMAC scores. After two years, VAS scores improved by 80% from pre to postembolization, and WOMAC scores improved by 85%. Additionally, the number of patients who used pain medication for OA reduced following GAE [59]. Of note, Torkian et al. also reported a 25.2% overall complication rate of GAE, the most common being transient cutaneous ischemia. Other adverse events reported include access-site hematomas, redness of the skin, and transient fever [59]. 

Despite the technical and clinical success of therapeutic embolization, the mechanism of action is not well understood. Okuno et al. have shown that targeted geniculate artery embolization resulted in statistically significant pain relief at 1-month, 4-month, and 12-month follow-up [17]. Midterm results demonstrated decreased WOMAC scores following geniculate artery embolization from an average baseline score of 43 +/− 8.3 to an average post-treatment score of 14 +/− 17 at 24 months [13]. Results also showed improved functionality and decreased pain symptoms of the knee for up to 4 years follow-up [13]. However, in these same patients, there was no significant improvement in the imaging appearance of their osteoarthritis on knee MRI as calculated by their WORMS scores nearly two years after treatment. Whole-Organ Magnetic Resonance Scoring (WORMS) suggests that although GAE results in a significant reduction in clinical symptoms, the imaging findings of osteoarthritis remain unchanged. Although the overall WORMS score did not improve in OA patients following GAE, the MRI appearance of synovitis was one imaging finding that did improve following GAE [13]. Similarly, a significant improvement in synovitis following GAE was observed in an interim analysis of GENESIS by Little et al. from WORMS [18]. Of note, Little et al. also found a significant deterioration in osteophytes and bone attrition from WORMS analysis, which was not previously reported. This was explained by the significantly higher median BMI (35) of the four patients who had a significant deterioration in osteophytes and bone attrition compared to the median BMI (25.2) of patients in the cohort included by Okuno et al. [13,18,60]. These four patients did, however, report improvement in the KOOS pain subscale. The overall results from Little et al. showed a significant decrease in mean visual analog scale (VAS) from 60 to 36 and 45 at 3 months and 1 year, respectively, and a significant improvement in the pain, other symptoms, function in sport and recreation, and knee-related quality of life subscales of the Knee Injury and Osteoarthritis Outcome Score (KOOS) questionnaire [18]. 

Early evidence shows that GAE is a straightforward procedure with less comorbidity relative to TKA that decreases pain via interruption of the inflammatory pathways and may also have a disease-modifying effect on the cartilage itself. These findings demonstrate the potential for neo-vessels as a target for disease-modifying treatment of OA as it relates to inflammation and synovitis [61,62,63]. Basic science, animal models, imaging studies, and early human trials demonstrate a link between synovitis, synovial angiogenesis, and OA. Our expanding knowledge of OA warrants further investigation of neo-vessels as a treatment target utilizing embolization therapy. In addition, prior literature has shown knee embolization to be a safe procedure with low complication rates, which has previously been used in other settings such as hemarthrosis [64,65,66,67,68]. 

Given the clinical need for effective non-surgical treatments of OA, evidence linking synovial neovascularization and the development of osteoarthritis, and clinical improvement in OA symptoms following GAE, future studies should seek to better understand the mechanisms involved in this burgeoning treatment modality. It has been nearly two decades since it was demonstrated during unanesthetized knee arthroscopy that “the anterior synovial tissues, fat pad, and capsule were exquisitely sensitive to the mechanical loading stimulus of the probe” [69], and in the interim, basic science research more accurately characterized the physiologic and pathologic basis of knee pain. Still, “further mechanisms by which transcatheter arterial embolization relieves patients’ symptoms remain obscure” [17], demanding further study in animal models and ultimately, in humans. 

## 4. Challenges and Recommendation

Understanding the inflammatory response to embolotherapy of OA will better guide treatment strategies. Although GAE has shown promise in treating intractable OA pain resistant to conservative management, the disease modification mechanism of embolization in OA remains poorly understood. The pathogenesis of OA results from a disturbance in the intra-articular microenvironment homeostasis toward a pro-inflammatory state associated with neovascularity and angiogenesis. These neo-vessels are suspected to be responsible for hyperalgesia in OA patients. However, the causality of these neo-vessels with respect to OA is not proven nor is the mechanism of action for GAE completely understood. The current animal models have only proven an association between microscopic neo-vessels and joints affected by OA. Identifying a similar type of macroscopic radiographically evident neo-vessel in an OA animal model would further elucidate the pathophysiology of this phenomenon in humans. Ultimately, continued study of intra-arterial joint embolization may lead to improved clinical outcomes for OA patients who do not respond to optimized non-operative therapy and are not considered surgical candidates. Geniculate artery embolization has shown promising early results for the management of mild to moderate OA of the knee, but expanded randomized controlled trials are needed to better evaluate its potential role in OA treatment. In a recent multicenter, randomized controlled trail, Bagla et al. showed that GAE significantly reduced pain and improved disability in patients with mild to moderate OA compared to patients of the sham group [58]. In addition, the Neovascularization Embolization for knee Osteoarthritis (NEO) trial is a randomized sham-controlled trial that is currently analyzing the safety of GAE for treating OA [70,71]. Correa et al. recently proposed the Genicular Artery embolization Using imipenem/Cilastatin vs. microspHere for knee Osteoarthritis (GAUCHO) trail, a randomized control trial, to compare impinem/cilstatin vs. microspheres in GAE for OA treatment [72]. While reducing patient pain is the chief concern in OA treatment, additional quantitative studies are needed to elucidate the downstream changes in biophysical factors and inflammatory cytokine cascade following GAE treatment. Additional human clinical trials coupled with expanded animal research on GAE will result in greater scientific understanding of the role of arterial embolization as a potential target treatment of osteoarthritis and may identify future targets for therapy. Appropriate indications and clinical application of embolization will be best informed via a thorough understanding of the effects of embolization on laboratory, histological and imaging correlates of OA. 

## Figures and Tables

**Figure 1 diagnostics-12-01403-f001:**
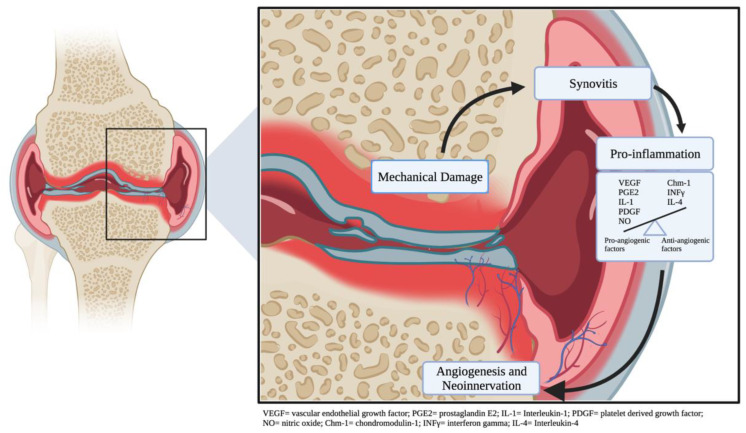
Understanding of the balance between pro and anti-angiogenic cytokines, inflammatory biomarkers.

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
