# Peer review of "Emerging Targets for the Treatment of Osteoarthritis: New Investigational Methods to Identify Neo-Vessels as Possible Targets for Embolization"

_diagnostics, 2022, doi:10.3390/diagnostics12061403_

Round 1

Reviewer 1 Report

Manuscript ID: diagnostics_1711024

Reviewer’s comments

This review article is well written with complete information. Below is a few suggestions.

  1. Provide one Table or Graph to illustrate mechanism of OA.
  2. Provide a case sample with MRI images to illustrate neovascularity around the OA knee before and after embolization.
  3. Which OA stage is eligible indication for IA treatment? Please describe.
  4. Okuno et al. reported improved WOMAC scores 2 years following IA treatment (Ref. #13). How about management of these patients after 2 years?
  5. Does IA treatment of OA knee have any side effects? Please describe.

Author Response

Dear sir or madam,

Thank you for giving us the opportunity to submit a revised draft of the manuscript titled “Emerging Targets for the Treatment of Osteoarthritis: A Literature Review and New Investigational Methods to Identify Neo-vessels as Possible Targets for Embolization” with submission reference number of “diagnostics-1711024”, to Diagnostics Journal.

We appreciate the time and effort that you and the reviewers have dedicated to providing valuable feedback on our manuscript. We are grateful to the reviewers for their insightful comments on the paper. We have been able to incorporate the changes to reflect all the suggestions provided.

We have uploaded a “highlighted” version of the manuscript with the changes based on the reviewers’ comments. We also uploaded a “clean” version of the manuscript as per your request.

Below, please find a point-by-point response to the reviewers’ comments and concerns.

Sincerely,

05/14/2022

Reviewer 1

This review article is well written with complete information. Below are a few suggestions.

Comment 1: Provide one Table or Graph to illustrate the mechanism of OA.

Response: Thanks for your insightful comment. We added an illustration in this regard as Figure 1.

Comment 2: Provide a case sample with MRI images to illustrate neovascularity around the OA knee before and after embolization.

Response: We appreciate the reviewer's comments. Since we started this trial recently in our own center, unfortunately we didn’t have MRI images after embolization.

Comment 3: Which OA stage is an eligible indication for IA treatment? Please describe.

Response: Thanks for the comment, we added that section. Genicular artery embolization (GAE) has been proposed as an additional, often supplementary method for management of mild to moderate OA of the knee. Mild to moderate OA treated by GAE using different embolic particles could generally be considered safe, with no reported serious complications.

Comment 4: Okuno et al. reported improved WOMAC scores 2 years following IA treatment (Ref. #13). How about the management of these patients after 2 years?

Response: Thanks for the comment. Okuno et al. reported follow-up of up to 4 years following GAE, and they reported improved functionality and decreased pain during this time. An additional sentence P7-Line 309-310 was added to reflect this. There were no direct WOMAC scores provided beyond the 2 year follow-up, however, so we were unable to include this information.

Comment 5: Does IA treatment of OA knee have any side effects? Please describe.

Response: The most common side effect of GAE is self-resolving transient cutaneous ischemia. Okuno et al. and Bagla et al. reported 57% and 65% skin discoloration, respectively, as an adverse event following GAE. This was clarified in P5-Line 235-239. Additionally, other side effects reported by Torkian et al. in a systematic review of the efficacy and safety of GAE include access-site hematomas, redness of the skin, and transient fevers. See P6-Line 301-303 for this included information.

Reviewer 2 Report

This review summarized the utility and mechanism of periarticular neovascular embolization for the treatment of OA. The comments are as below:

1.P2-Line72: The predicate should be plural.

2.Notice the format and remove the manual line breaks, such as P3-Line123, P3-Line132, 150, and so on.

3.Attention should be paid to the standardization of sentence juxtaposition or punctuation, suan as P3:Line139-140, P3:Lne143-144 and so on.

4.P3-Line141,142: The coordinate elements in not only,but also sentence patterns are not standardized, and Swap the position of not only and possess.

5.P6-Line282: Why is most capitalized?

6.References are not uniform in format.

Author Response

Dear sir or madam,

Thank you for giving us the opportunity to submit a revised draft of the manuscript titled “Emerging Targets for the Treatment of Osteoarthritis: A Literature Review and New Investigational Methods to Identify Neo-vessels as Possible Targets for Embolization” with submission reference number of “diagnostics-1711024”, to Diagnostics Journal.

We appreciate the time and effort that you and the reviewers have dedicated to providing valuable feedback on our manuscript. We are grateful to the reviewers for their insightful comments on the paper. We have been able to incorporate the changes to reflect all the suggestions provided.

We have uploaded a “highlighted” version of the manuscript with the changes based on the reviewers’ comments. We also uploaded a “clean” version of the manuscript as per your request.

Below, please find a point-by-point response to the reviewers’ comments and concerns.

Sincerely,

05/14/2022

Reviewer 2

Comment 1: P2-Line72: The predicate should be plural.                                                                 Response: This change has been made. “Has” was changed to “have.”

Comment 2: Notice the format and remove the manual line breaks, such as P3-Line123, P3-Line132, 150, and so on.                                                                                                                                

Response: Thanks and all line breaks and manual formatting have been updated. Line breaks from P3-Line123, P3-Line125, P3-Line132, P3-Line150, P8-Line377 have been removed.

Comment 3: Attention should be paid to the standardization of sentence juxtaposition or punctuation, suan as P3:Line139-140, P3:Lne143-144 and so on.                                                                  

Response: These changes have been made. P3-Line136-137: A comma was added after “with pathological vascularization” and a comma was removed after “however”.  P3-Line 141: A period was added to correct punctuation.

Comment 4: P3-Line 141,142: The coordinate elements in not only,but also sentence patterns are not standardized, and Swap the position of not only and possess.                                                    

Response: The changes have been made. P3-Line 138-139. The positions of not only and possess were swapped.

Comment 5: P6-Line282: Why is most capitalized?                                                                   

Response: MOST is an abbreviation for Multicenter Osteoarthritis Study, so clarification was added to P6-Line283. MOST was also un-bolded.

Comment 6: References are not uniform in format.                                                                    

 Response: References have been updated so that they are in accordance with the Reference List and Citations Style Guide for MDPI Journals.

Round 2

Reviewer 2 Report

This manuscript have solved all raised questions and can be published by the journal.